# Nanoscale zero-field electron spin resonance spectroscopy

Fei Kong[1,2,3], Pengju Zhao[1,3], Xiangyu Ye[1,3], Zhecheng Wang[1,3], Zhuoyang Qin[1,3], Pei Yu[1,3], Jihu Su[1,2,3], Fazhan Shi[1,2,3] & Jiangfeng Du[1,2,3]

Electron spin resonance (ESR) spectroscopy has broad applications in physics, chemistry, and biology. As a complementary tool, zero-field ESR (ZF-ESR) spectroscopy has been proposed for decades and shown its own benefits for investigating the electron fine and hyperfine interaction. However, the ZF-ESR method has been rarely used due to the low sensitivity and the requirement of much larger samples than conventional ESR. In this work, we present a method for deploying ZF-ESR spectroscopy at the nanoscale by using a highly sensitive quantum sensor, the nitrogen vacancy center in diamond. We also measure the nanoscale ZF-ESR spectrum of a few P1 centers in diamond, and show that the hyperfine coupling constant can be directly extracted from the spectrum. This method opens the door to practical applications of ZF-ESR spectroscopy, such as investigation of the structure and polarity information in spin-modified organic and biological systems.

[1] CAS Key Laboratory of Microscale Magnetic Resonance and Department of Modern Physics, University of Science and Technology of China (USTC), 230026 Hefei, China. [2] Hefei National Laboratory for Physical Sciences at the Microscale, USTC, 230026 Hefei, China. [3] Synergetic Innovation Center of Quantum Information and Quantum Physics, USTC, 230026 Hefei, China. These authors contributed equally: Fei Kong, Pengju Zhao. Correspondence and requests for materials should be addressed to F.S. (email: fzshi@ustc.edu.cn) or to J.D. (email: djf@ustc.edu.cn)

Electron spin resonance (ESR) spectroscopy is a method for studying paramagnetic targets, including metal complexes and organic radicals. Combining with site-directed spin-labeling of radicals, ESR has been widely used to study basic molecular mechanisms in biological systems[1], such as structure[2], dynamics[3], and polarity[4]. This information is derived from the electron fine and hyperfine interaction, which can be obtained from the ESR spectra with accuracy limited by the inhomogeneous line broadening. The line broadening of powder spectra, induced by the magnetic inequivalence of otherwise-identical spins, can be partly removed in high-field ESR (HF-ESR) experiment[5]. A complete removal can be achieved by zero-field ESR (ZF-ESR) spectroscopy[6–9] because the energy levels of a spin system are no longer orientation dependent in the absence of a Zeeman term. Although the ZF-ESR spectroscopy provides a straightforward method for investigating the intrinsic interactions, it has rare applications due to the much lower sensitivity compared with the conventional ESR (henceforth, ESR refers to non-zero field ESR).

Nitrogen vacancy (NV) centers in diamond can be used as magnetic quantum sensors to significantly improve the sensitivity[10–12,13]. NV center-based ESR has been demonstrated using both double electron–electron resonance (DEER)[14,15,16] and cross-polarization methods[17,18], and detection of single electron spins has been realized[19,20]. However, these methods cannot be directly used in zero field because of the complication of manipulating target spins in zero field.

Here we propose a different method, which does not require any manipulation of target spins, and allows for detecting nanoscale ZF-ESR signals. It is achieved by precisely tuning the energy levels of NV centers in dressed states to be resonant with target spins. We also demonstrate it by detecting the ZF-ESR spectrum of a few P1 centers residing at a distance of less than 15 nanometers from the NV center. By analyzing the resonance lines, the intrinsic interactions, such as hyperfine interaction, can be directly resolved. Moreover, we show that the nanoscale ZF-ESR can perform significantly better than nanoscale ESR without loss of sensitivity.

## Results

**Scheme of zero-field detection using an NV spin.** The basic essence of our method is shown in Fig. 1a, where the sensors are shallow NV centers embedded in a bulk diamond. The diamond is adhered on a coplanar waveguide, and observed by a home-built confocal microscope from the other side. As illustrated in Fig. 1b, the NV centers are surrounded by a spin mixture consisting of target spins and background spins. The target spins can be either defects inside the diamond or paramagnetic targets on the diamond surface. The background spins are dangling bonds on the diamond surface. The NV center is a kind of color defect in diamond consisting of a substitutional nitrogen atom and an adjacent vacancy (see Fig. 1c). The electrons around the defect form an effective electron spin with a spin triplet ground state ($S = 1$) and a zero-field splitting of $D = 2\pi \times 2.87$ GHz[21]. The NV electron spin can be polarized into the $|m_s = 0\rangle$ state ($|0\rangle$, henceforth) by 532 nm laser excitation. The photoluminescence (PL) rate of NV center is state-dependent, i.e., $|0\rangle$ is "brighter" than $|\pm 1\rangle$, then the populations of the states can be directly read out by the photoluminescence rate. Furthermore, the spin transition between $|0\rangle$ and $|\pm 1\rangle$ can be driven by the microwave. As shown in Fig. 1d, under resonant microwave ($B_1 \cos ft$, $f = D$) radiation, the NV electron spin can be driven to the dressed states, which are

$$|-1\rangle_d = \tfrac{1}{2}|1\rangle - \tfrac{1}{\sqrt{2}}|0\rangle + \tfrac{1}{2}|-1\rangle,$$
$$|0\rangle_d = -\tfrac{1}{\sqrt{2}}|1\rangle + \tfrac{1}{\sqrt{2}}|-1\rangle, \qquad (1)$$
$$|1\rangle_d = \tfrac{1}{2}|1\rangle + \tfrac{1}{\sqrt{2}}|0\rangle + \tfrac{1}{2}|-1\rangle,$$

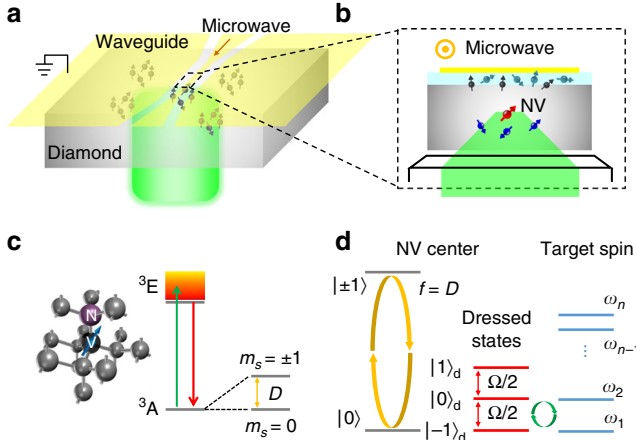

**Fig. 1** Essence of the method for ZF-ESR spectroscopy at the nanoscale. **a** Sketch of the experimental setup. The diamond with shallow NV centers is adhered on a coplanar waveguide, which can radiate microwave from the central gold wire. **b** Sectional view of the central area of the setup. The red, blue (turquoise), and black arrows denote the NV center, inner (outer) target spins, and background spins, respectively. **c** Structure and energy levels of the NV center. The NV center can be excited from the ground states ³A to the excited states ³E by a laser pulse, and then decays back to ³A with emission of photoluminescence. The spin transition between the ground states $|m_s = 0\rangle$ and $|m_s = \pm 1\rangle$ with a zero-field splitting $D$ can be driven by microwave pulses. **d** Dressed states of the NV center being resonant with the target spin. The ground states ($|0\rangle$, $|\pm 1\rangle$) can be driven to the dressed states ($|0\rangle_d$, $|\pm 1\rangle_d$) by a resonant microwave pulse with frequency $f$. The energy level splitting in dress states can be tuned by the driving power $\Omega$. The energy levels of the target spin are denoted as $\omega_j$ ($j = 1,2,...,n$). When $\Omega/2$ matches the energy level splitting of the target spin, the flip-flop process between the NV center and the target spin occurs

in the rotating reference frame with eigenenergies $-\Omega/2$, 0, and $\Omega/2$, respectively. Here, $\Omega$ is the driving power valued by the corresponding Rabi frequency with $\Omega = \gamma_{NV} B_{1,\perp}$ proportional to the perpendicular amplitude of the driving microwave, where $\gamma_{NV} = -2\pi \times 2.803$ MHz/G is the gyromagnetic ratio of the NV electron spin. Similar to the Hartmann–Hahn double resonance[22,23], the dipole–dipole interaction between the NV center and the target spins will induce a flip-flop process at the resonance condition

$$\Omega = 2\Delta\omega_{ij}, \qquad (2)$$

where $\Delta\omega_{ij} = \omega_j - \omega_i$ ($i < j$) is the energy level splitting of the target spins corresponding to an allowed magnetic dipole transition. If the NV center is initially polarized to one of the dressed states given by Eq. (1), the polarization will be transferred from the NV center to the target spins, then the populations of dressed states will change. Therefore, a resonance spectrum of the target spins can be measured by sweeping the driving power and measuring the populations of the dressed states.

In zero magnetic field, all the Zeeman terms vanish and the energy level structure of the target spins is determined solely by the intrinsic spin–spin interactions. To see how these interactions can be resolved by ZF-ESR, we take a free radical with hyperfine interaction between the electron spin **S** and the nuclear spin **I** as an example. For clarity, here we deal with $S = 1/2$ and $I = 1/2$ (for $I = 1$ nuclei, see Supplementary Note 5). The Hamiltonian of the target spin can be written as[24]

$$H_0 = \mathbf{S} \cdot \mathbb{A} \cdot \mathbf{I}, \qquad (3)$$

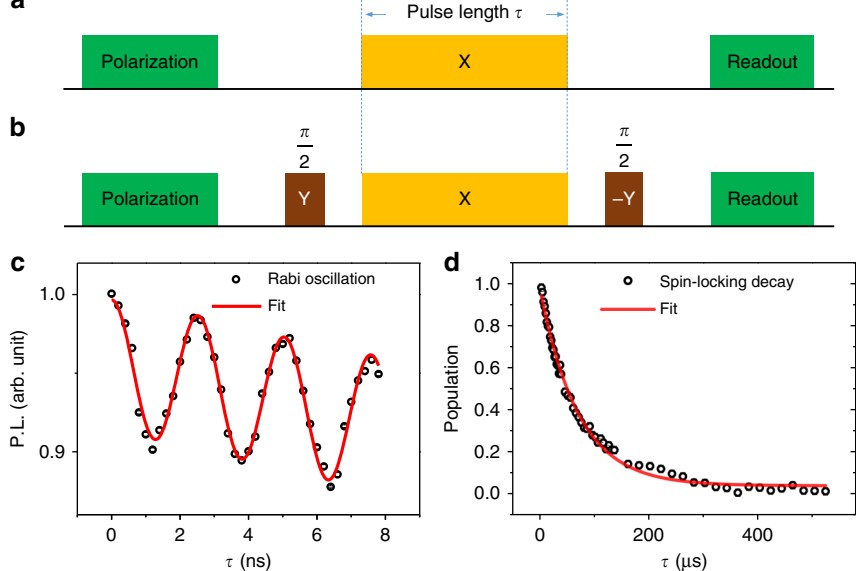

**Fig. 2** State evolution of the NV center with continuous driving. **a** Pulse sequence for Rabi measurement. Green boxes denote the laser pulses used to polarize and read out the NV center. Orange box denotes the microwave pulse used to drive the NV states. **b** Pulse sequence for spin-locking measurement. Two additional $\pi/2$ pulses with 90° phase difference (brown boxes) are added before and after the continuous driving. **c** Measured Rabi oscillation. A modulated sine fit gives the Rabi frequency of 396 MHz. Error bars indicate ±1 standard error of the mean (s.e.m), which is induced by the photon shot noise. Here, error bars are smaller than the symbols, as the sequence is repeated 40 thousand times to get enough photons. **d** Measured spin-locking relaxation. The driving power $\Omega = 148$ MHz. An exponential decay fit gives a relaxation time of $70 \pm 2$ μs. Error bars (±1 s.e.m.) are smaller than the symbols. The sequence is repeated 60 thousand times

where $\mathbb{A}$ is the hyperfine tensor, which is diagonal

$$\mathbb{A} = \begin{pmatrix} A_{xx} & & \\ & A_{yy} & \\ & & A_{zz} \end{pmatrix} \quad (4)$$

in the principal axis frame. The eigenstates of Eq. (3) are

$$\begin{aligned} |\phi_1\rangle &= \tfrac{1}{\sqrt{2}}(|\uparrow\downarrow\rangle - |\downarrow\uparrow\rangle), \\ |\phi_2\rangle &= \tfrac{1}{\sqrt{2}}(|\uparrow\downarrow\rangle + |\downarrow\uparrow\rangle), \\ |\phi_3\rangle &= \tfrac{1}{\sqrt{2}}(|\uparrow\uparrow\rangle - |\downarrow\downarrow\rangle), \\ |\phi_4\rangle &= \tfrac{1}{\sqrt{2}}(|\uparrow\uparrow\rangle + |\downarrow\downarrow\rangle), \end{aligned} \quad (5)$$

where $|\uparrow\rangle$ and $|\downarrow\rangle$ denote spin-up and spin-down, respectively, with the electron spin on the left and the nuclear spin on the right. The corresponding eigenenergies are

$$\begin{aligned} \omega_1 &= \tfrac{1}{4}(-A_{xx} - A_{yy} - A_{zz}), \\ \omega_2 &= \tfrac{1}{4}(A_{xx} + A_{yy} - A_{zz}), \\ \omega_3 &= \tfrac{1}{4}(-A_{xx} + A_{yy} + A_{zz}), \\ \omega_4 &= \tfrac{1}{4}(A_{xx} - A_{yy} + A_{zz}). \end{aligned} \quad (6)$$

Six allowed magnetic dipole transitions between these states are expected with frequencies

$$\begin{aligned} \Delta\omega_{12} &= \tfrac{1}{2}|A_{xx} + A_{yy}|, \quad \Delta\omega_{34} = \tfrac{1}{2}|A_{xx} - A_{yy}|, \\ \Delta\omega_{13} &= \tfrac{1}{2}|A_{yy} + A_{zz}|, \quad \Delta\omega_{24} = \tfrac{1}{2}|A_{yy} - A_{zz}|, \\ \Delta\omega_{14} &= \tfrac{1}{2}|A_{xx} + A_{zz}|, \quad \Delta\omega_{23} = \tfrac{1}{2}|A_{xx} - A_{zz}|. \end{aligned} \quad (7)$$

Note that a rotational transformation does not change the eigenenergies, which means that these transition frequencies are independent of the orientation of the radicals, while the transition

speed determined by the dipole–dipole interaction between the NV center and the radicals is orientation-dependent. A detailed discussion can be found in Supplementary Note 1.

**Experimental demonstration.** We perform the measurement on a similar electron-nuclear system named a P1 center, which is another kind of defect in diamond consisting of only a sub-stitutional nitrogen atom. For a $^{15}$N P1 center, the principal values of the hyperfine tensor are $A_{xx} = A_{yy} = A_\perp = 114$ MHz and $A_{zz} = 159.9$ MHz[25]. Due to the Jahn–Teller effect, one of the nitrogen–carbon bonds is distorted, and the orientation of a P1 center can change between the four kinds of N–C bonds[26]. The NV and P1 centers are generated by ion implantation. To make the P1 centers close enough to the NV centers, the diamond we used is implanted with high dose of nitrogen ions (see "Meth-ods"). So the experiment is performed on an ensemble of NV centers with each NV center surrounded by dozens of P1 centers. As demonstrated by Hall et al.[18], the NV center is a highly local sensor. Although ensembles of NV and P1 centers are involved, the response of each NV center is dominated by only those P1 centers residing in its nanoscale detection area (a quantitative discussion is given below). Therefore, each NV center is an iso-lated sensor, and the detected ZF-ESR spectrum is the average spectrum of P1 centers in each nanoscale region. Since the resonance frequencies of different P1 centers are the same, using ensembles of NV centers can speed up the measurement.

Unlike the frequency-sweeping method[14,15,16,19,20], where the microwave frequency can be easily tuned, the microwave power is limited by the microwave amplifier and the transform efficiency of the waveguide. Here, by using a proper designed coplanar waveguide, a Rabi frequency up to ~400 MHz was realized (see Fig. 2a, c), enough to capture all the interaction information of P1 centers. We note that by further employing a high-power microwave amplifier and shaped pulses, Rabi frequencies ~1 GHz are possible[27], which is theoretically limited by the zero-field splitting of NV centers. Then, this method is applicable to more

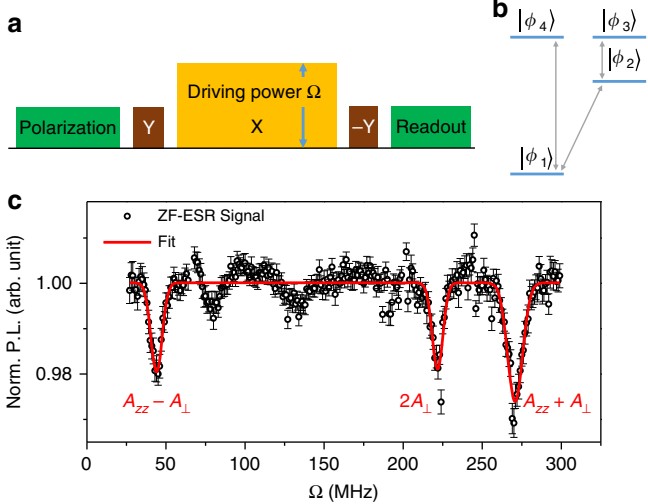

**Fig. 3** Measurement of the ZF-ESR spectrum of P1 centers. **a** Revised spin-locking sequence. Instead of sweeping the length of the driving pulse (orange boxes), here we sweep the power of the driving pulse. **b** Energy levels of $^{15}$N P1 centers. Due to the degeneracy of the upper two energy levels, the magnetic dipole transitions denoted by gray arrows have only three kinds of frequencies. **c** Measured ZF-ESR spectrum of $^{15}$N P1 centers. The P.L. is normalized by the amplitude of Rabi oscillation. The length of driving pulse is $\tau = 10\,\mu s$. The circle points are experimental results while the solid line is a three-Gaussian-peak fitting. The measurement sequence is repeated several million times, and error bars indicate ±1 s.e.m.

high-spin paramagnetic targets. On the other hand, the lower limit of the Rabi frequency is bounded by the dephasing time of NV centers. The NV centers have four kinds of directions, so the $B_{1,\perp}$ on different NV sites may be different, which induces an extra sine modulation on the Rabi oscillation. Comparing with the Rabi oscillation, where the NV state is rotating between $|0\rangle$ and $|\pm 1\rangle$, the NV state is locked in $|-1\rangle_d$ by applying a $\pi/2$ pulse with 90° phase difference before the continuous driving (see "Methods" and Fig. 2b). The experimental result in Fig. 2d shows that the NV state is successfully locked with a relaxation time of $T_{1\rho} = 70 \pm 2\,\mu s$, which is much longer than the dephasing time of $T_2^* \sim 0.1\,\mu s$.

The ZF-ESR spectrum is obtained by sweeping the driving power while fixing the driving length $\tau = 10\,\mu s$, as illustrated in Fig. 3a. Since the hyperfine tensor $\mathbb{A}$ of P1 centers is cylindrically symmetrical, the eigenstate $|\phi_3\rangle$ and $|\phi_4\rangle$ are degenerate, thus only three magnetic dipole transitions are observable (see Fig. 3b). Figure 3c gives the measured spectrum with three characteristic peaks at $44.0 \pm 3.6$, $221.7 \pm 3.4$, and $270.8 \pm 4.8$ MHz, where the error bars are given by half the linewidths of the Gaussian peaks. It seems like there are other peaks at ~80 and ~130 MHz. The reason is unknown yet. A possible source of these extra signals is other kinds of defects, due to the numerous electron spin defects in diamond[28]. As the spin-locking relaxation time $T_{1\rho}$ depends on the driving power $\Omega$, the baseline of the original ZF-ESR spectrum is uneven (Supplementary Note 3), which has been calibrated to unity as shown in Fig. 3c. Combining with Eqs. (2) and (7), the principal values of the hyperfine tensor can be directly obtained from any two peaks or a least-squares fitting of all the three peaks. The least-squares fitting gives $A_\perp^{exp} = 110.7 \pm 3.5$ MHz and $A_{zz}^{exp} = 155.0 \pm 7.1$ MHz, consistent with the hyperfine coupling of $^{15}$N P1 center. We have also measured the ZF-ESR spectrum of $^{14}$N P1 center, which can be found in Supplementary Note 5.

The energy level shifts induced by the environmental fluctuations on either the NV center or the P1 center will destroy the energy exchange between them, which acts to broaden the resonance frequencies, as characterized by the transverse spin relaxation rate, $\Gamma_{tot} = \Gamma_{NV} + \Gamma_{P1}$. The transverse spin relaxation of the NV center can be significantly suppressed by continuously driving[29], so $\Gamma_{tot} \approx \Gamma_{P1} = 1/T_{2,P1}^*$. The dephasing time of the P1 centers can be estimated by the dephasing time of the NV center, which is ~0.1 μs, because of the similar environment of the NV and P1 centers. Thus, the linewidth induced by the dephasing is $\Delta\Omega \sim 1/(\pi T_{2,P1}^*) \sim 3.2$ MHz[17], not enough to explain the observed linewidth of 7 ~ 10 MHz. Besides, the fluctuations of the driving power will also induce the broadening of zero-field lines. It was recorded and calibrated during the measurement with fluctuation <0.5% (Supplementary Note 4). Moreover, the residual static magnetic field (mainly contributed by the geomagnetic field) will induce line splitting of ~2.3 MHz (Supplementary Note 4), which will behave as line broadening if the line splitting is smaller than the linewidth. The residual field can be further avoided by placing the setup in a mu-metal box or actively compensating with an electromagnetic field.

To confirm that the detected spectrum is indeed dominated by the P1 centers in a nanoscale detection area, we have calculated the signal induced by all the P1 centers residing at a distance of >15 nm from the NV center, which is ≤0.29% (see "Methods"). While the signal contrast in Fig. 3c is ~3%, indicates more than 90% of the signal is attributable to the P1 centers residing in the detection area with radius of 15 nm. The mean number of P1 centers in this detection area is ~4. Note that it is a conservative estimate, as the peaks in Fig. 3c are broadened by some additional effects which will lower the signal contrast.

**Comparison to nanoscale ESR**. In some practical nanoscale applications, such as detection of nitroxide radicals dispersed on the diamond surface, the NV center-based ZF-ESR can outperform NV center-based ESR (here we take DEER as an example) for several reasons, including the following. First, the ZF-ESR method, which does not require an external magnet or manipulation of the target spins (see Fig. 4a, b), will be easily used. Second, unlike the inner P1 centers, which are restricted by the diamond lattice and have only four orientations, the external nitroxides are much more disordered and can be oriented in any direction[16,20]. The simulated spectra with different orientations of nitroxide are given in Fig. 4c, d. Since the DEER spectrum is orientation-dependent, the peaks will be averaged if differently oriented nitroxides are included. For conventional induction-based ESR, the statistical average of the orientations over a large ensembles of spins induces an inhomogeneously broadened powder spectrum, where the hyperfine values can still be extracted even with limited accuracy. But for nanoscale ESR, the detected spins are much fewer (~$10^1$), then the average spectrum (see Fig. 4e) still has a certain randomness. Without prior knowledge of the orientations of each nitroxide, extraction of the hyperfine values will be nearly impossible. However, for the ZF-ESR spectrum, different orientations will lead to different peak intensities rather than peak positions, so the characteristic peaks remain clear in the ZF-ESR spectrum (see Fig. 4f) all the time. Third, there exists a paramagnetic background, which consists of dangling bonds on the diamond surface with g factor similar to free radicals[30–32]. Due to the dense distribution and the short correlation time[30,33,34], the background spins always behave as a broad central peak in the DEER spectrum (see Fig. 4e and Supplementary Note 2), which can be hardly distinguished from the nitroxides. While in zero field, the background peak will move to the zero frequency and only a characterless low-frequency signal

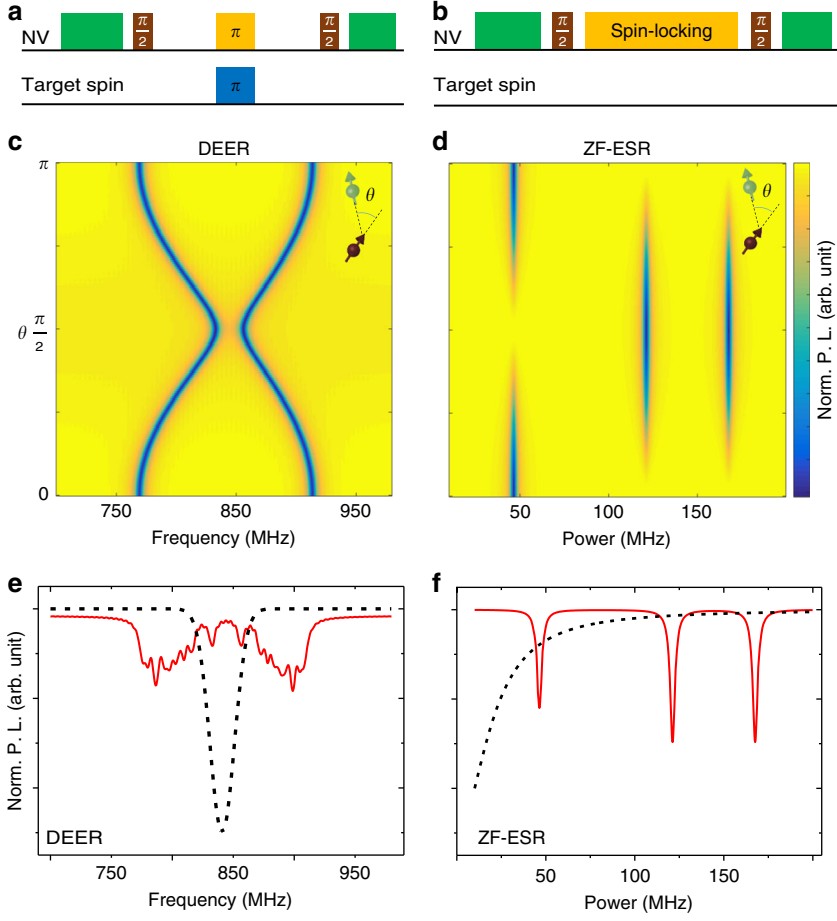

**Fig. 4** Comparison between DEER method and ZF-ESR method. **a** Pulse sequence for DEER measurement. The frequency of the pulse applied on the target spin (blue box) is swept. **b** Pulse sequence for ZF-ESR measurement. The power of the driving pulse (orange box) is swept. **c, d** Simulated DEER and ZF-ESR spectra of differently oriented nitroxides. $\theta$ is the angle between the principal axis of the nitroxide and the $N$–$V$ axis. The magnetic field for DEER measurement is 300G with direction parallel to the $N$–$V$ axis. The principal values of the hyperfine tensor are $A_{xx} = A_{yy} = 23.2$ MHz and $A_{zz} = 144.4$ MHz. **e, f** Schematic diagram of the background signal. The red solid and dark dashed lines indicate the signal of the target spins and the background spins, respectively. Here, the target spins are 10 randomly oriented nitroxides

can be observed (see Fig. 4f and Supplementary Note 2), thus the fingerprint-like ZF-ESR signal of target spins can be naturally distinguished.

On the other hand, unlike the conventional induction-based ESR, where the signal intensity is proportional to the polarization of the electron spins (i.e., the magnetic field), the signal of NV center-based ESR is induced by the statistically polarized electron spins. So, the sensitivity of this nanoscale ESR does not depend on the magnetic field. Specifically, the nanoscale detection is based on dipole–dipole coupling between the probe and the target and limited by the coherence properties of the probe. For DEER measurement, only $zz$ coupling contributes to the signal, and the sensitivity is limited by the $T_2$ of the NV center[12]. For our ZF-ESR measurement, all the $zx$, $zy$, and $zz$ couplings contribute to the signal (detail in Supplementary Note 1), and the sensitivity is limited by the $T_{1\rho}$ of the NV center, which is usually much longer than $T_2$. Therefore, the sensitivity of our nanoscale ZF-ESR is comparable with and even potentially better than the nanoscale ESR.

## Discussion

In conclusion, we have presented a method for measuring the ZF-ESR spectrum of nanoscale paramagnetic samples, via continuously tuning the energy levels of NV centers in dressed states. The significant improvement of the sensitivity of ZF-ESR removes

the biggest obstacle to applications of ZF-ESR spectroscopy. The nanoscale ZF-ESR can also outperform the previous nanoscale ESR. The method is demonstrated by successful measurement of the ZF-ESR spectrum of a few P1 centers at the nanoscale. Unlike the conventional ESR spectrum which is complicated due to the orientation-dependent resonance frequencies, our method of measurement in zero field can unambiguously give a characteristic resonance spectrum determined solely by the intrinsic interactions of the target spins, and thus the interaction structure can be directly resolved. It was found that the hyperfine interaction has a strong relation to the polarity profiles and hydrogen bonding effects of molecules[4], so this method can be used further to analyze the micro-environment information of molecules. Although this work is focused on the electron–nuclear hyperfine interaction, we note that the key element of the technique is the direct measurement of the energy level structure of paramagnetic targets, regardless of the form of interaction. This technique can also be used to analyze the electron–electron spin interaction, which contains the structure information of molecules[2].

## Methods

**Experimental setup**. The setup is based on a home-built confocal microscopy. A temperature-stabilized diode laser (CNI MGL-III-532) is used for the polarization and readout of the NV centers. The photoluminescence is detected by an Avalanche photodiode (Perkin Elmer SPCM-AQRH-14) through a ×60, 0.7 NA objective (Olympus LUCPLFLN). The driving microwave is generated by an

arbitrary waveform generator (Agilent M8190a) and amplified by a microwave amplifier (Mini-circuits ZHL-16W-43+). The diamond is obtained commercially, 100-oriented, and electronic-grade with a thickness of 500 μm. The NV and P1 centers were created by implantation of 5 keV $^{15}N^+$ ions with dose of $5.5 \times 10^{11}$ cm$^{-2}$. The mean spacing of N atoms is 13.5 nm. The production efficiency of NV centers is ~1%, and then the mean spacing of NV centers is ~135 nm. The mean depth of N atom is estimated to be $8 \pm 3.2$ nm by SRIM, while the actual depth may be larger[35].

**Detection area**. Since the NV and P1 centers are created by the implantation of N$^+$ ions, we suppose both of them are located in a thin layer and only the transverse distance is considered. It is reasonable when the transverse distance is much larger than the longitudinal straggling. Here, we consider the sum of the signal contributed by all the P1 centers out of the detection area, which can be given by (see Supplementary Note 1)

$$
\begin{aligned}
S^{\text{sum}}(r > r_0) &= \sum_{r > r_0} \frac{C_0^2 \langle \eta^2 \rangle \tau}{8 \Gamma_{P1} r^6} \\
&= \int_{r_0}^{\infty} \frac{C_0^2 \langle \eta^2 \rangle \tau}{8 \Gamma_{P1} r^6} \cdot \sigma \cdot 2\pi r \, dr \\
&= \frac{\pi \sigma C_0^2 \langle \eta^2 \rangle \tau}{16 \Gamma_{P1} r_0^4},
\end{aligned}
\tag{8}
$$

where $\sigma = 5.5 \times 10^{11}$ cm$^{-2}$ is the area density of P1 centers, $C_0 = 2\pi \times 52$ MHz nm$^3$ is the dipolar coupling constant, $\eta$ is the coefficient of dipole–dipole coupling depending on both the direction of the spatial separation $\mathbf{r}$ and the P1 center. After averaging over all the possible directions, $\langle \eta^2 \rangle$ is calculated to be 5/4 or 3/4 corresponding to the left/right or middle peak in Fig. 3c. Other parameters in this experiment are $\Gamma_{P1} = 10$ MHz and $\tau = 10$ μs. Hence, the sum of signals contributed by those P1 centers out of the detection area decreases dramatically with the radius of the area. For example, if $r_0 = 15$ nm, then $S^{\text{sum}}$ can be calculated to be 0.29%, 0.17%, and 0.29% for the left, middle, and right peaks, respectively.

**Spin locking in zero field**. The Hamiltonian of NV center in zero magnetic field is

$$
H_0 = D S_z^2,
\tag{9}
$$

where $\mathbf{S}(S = 1)$ is the NV electron spin operators, $D = 2\pi \times 2.87$ GHz is the zero-field splitting. The NV center can be driven by the microwave of the form

$$
H_1 = \Omega \cos(ft + \phi) S_x,
\tag{10}
$$

where $\Omega$, $f$, and $\phi$ are the power, frequency, and phase of the microwave, respectively, $f = D$ in the resonance condition. In rotating reference frame, the Hamiltonian becomes

$$
\begin{aligned}
H^{\text{rot}} &= e^{iDS_z^2 t}(H_0 + H_1)e^{-iDS_z^2 t} - D S_z^2 \\
&= \frac{\Omega}{2\sqrt{2}} \begin{pmatrix} 0 & e^{-i\phi} & 0 \\ e^{i\varphi} & 0 & e^{i\phi} \\ 0 & e^{-i\phi} & 0 \end{pmatrix},
\end{aligned}
\tag{11}
$$

where the high frequency items are ignored. By defining $\phi = 0$, $\pi/2$, and $-\pi/2$ for $x$, $y$, and $-y$ phase, respectively, we can write

$$
\begin{aligned}
H_x^{\text{rot}} &= \frac{\Omega}{2} S_x, \\
H_y^{\text{rot}} &= \frac{\Omega}{2} S_{yy}, \\
H_{-y}^{\text{rot}} &= -\frac{\Omega}{2} S_{yy}.
\end{aligned}
\tag{12}
$$

Note here

$$
S_{yy} = \frac{1}{\sqrt{2}} \begin{pmatrix} 0 & -i & 0 \\ i & 0 & i \\ 0 & -i & 0 \end{pmatrix}
\tag{13}
$$

is slightly different from $S_y$. The eigenstates and eigenenergies of $H_x^{\text{rot}}$ can be directly calculated as given in the main text Eq. (1). The spin-locking sequence is performed as following:

First, the initial state of the NV center is polarized to $|\psi_0\rangle = |0\rangle$ by a laser pulse. Then, a $y$-phase $\pi/2$ pulse with power $\Omega_0$ and length $t_{\pi/2} = 1/(4\Omega_0)$ is applied and the NV state is prepared to

$$
\begin{aligned}
|\psi_1\rangle &= e^{-i 2\pi H_y^{\text{rot}} t_{\pi/2}} |\psi_0\rangle \\
&= \frac{1}{2}|1\rangle - \frac{1}{\sqrt{2}}|0\rangle + \frac{1}{2}|-1\rangle \equiv |-1\rangle_d.
\end{aligned}
\tag{14}
$$

Second, a $x$-phase pulse with power $\Omega$ and length $\tau$ is applied. Since $|\psi_1\rangle$ is an eigenstate of $H_x^{\text{rot}}$, the state will be locked in $|\psi_1\rangle$. However, it can be transferred to other dressed states by $T_{1\rho}$ relaxation or coupling with nearby P1 centers.

Supposing the transition probability is $P$, then the state becomes

$$
\rho_2 = (1 - P)|-1\rangle_d \langle -1| + P(p_1|0\rangle_d \langle 0| + p_2|1\rangle_d \langle 1|).
\tag{15}
$$

where $p_1 + p_2 = 1$.

Finally, a $-y$-phase $\pi/2$ pulse with power $\Omega_0$ and length $t_{\pi/2} = 1/(4\Omega_0)$ is applied. The state $|-1\rangle_d$ will be rotated back to $|0\rangle$, while $|0\rangle_d$ remains unchanged and $|1\rangle_d$ will be rotated to $|0\rangle_d$. Therefore, the final state is

$$
\rho_3 = (1 - P)|0\rangle\langle 0| + P(|1\rangle\langle 1| - |1\rangle\langle -1| - |-1\rangle\langle 1| + |-1\rangle\langle -1|),
\tag{16}
$$

with the normalized photoluminescence of $1 - P$.

**Data availability**. Data supporting the findings of this study are available within the article and its Supplementary Information file, and from the corresponding authors on reasonable request.

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

## Acknowledgements

This work was supported by the 973 Program (Grants No. 2016YFA0502400 and 2013CB921800), the National Natural Science Foundation of China (Grants No. 11227901, 91636217, 11722544, 31470835, and 81788104), the CAS (Grants No. XDB01030400, QYZDY-SSW-SLH004, and YIPA2015370), the CEBioM, and the Fundamental Research Funds for the Central Universities (WK2340000064).

## Author contributions

J.D. supervised the entire project. J.D., F.K., and F.S. proposed the idea and designed the experiments. F.K. and P.Z. prepared the setup and performed the experiments. X.Y. and P.Y. prepared the diamond sample. F.K. carried out the calculations. F.K., F.S., P.Z., and J.D. wrote the manuscript. All authors analyzed the data, discussed the results, and commented on the manuscript.

## Additional information

**Competing interests:** The authors declare no competing interests.

