## [Peer Review File · Nature Communications]

Reviewers' comments:

Reviewer #1 (Remarks to the Author):

This manuscript demonstrates zero-field EPR spectroscopy on P1 centers implanted into diamond by using nearby NV centers for sensitive detection. The experiment is based on dipole-induced polarization transfer between the NV observer and the observed P1 centers. The authors rightly argue that nanoscale EPR at zero field (or earth field) has advantages compared to such experiments in the presence of a strong magnetic field, which breaks symmetry and thus introduces orientation dependence of the transition frequencies. This is a highly significant advance in EPR spectroscopy on the nanoscale, which will probably have great influence on further development of this exciting field. It is also a very elegant use of the magnetic sensor properties of NV centers. The results and ideas are thus of sufficiently broad interest for publication in Nature Communications.

In general, the manuscript is well written, clear, and concise. The conclusions are fully supported by the experimental evidence. Yet, a few points in the discussion should be improved, there are minor mistakes, and in some places the English needs to be improved before a non-specialist can do further language editing (see below). Therefore, minor revision is required.

Details:

1. The specialist will understand from your wording that the product of $T_1\rho$ with the dipole-dipole coupling between the NV center and the observed species determines whether the experiment is feasible. For a broad readership, this should be made explicit and, given that you know the dependence of $T_1\rho$ on driving power, you should provide an estimate for the maximum distance between the NV center and the observe species where the experiment works.
2. It is true that you use a kind of cross-polarization technique and it may be appropriate to cite the Hartmann-Hahn paper. However, in your case polarization is transferred from a dressed species to a "bare" species, which is the type of transfer first shown in the NOVEL dynamic nuclear polarization experiment (A. Henstra, P. Dirksen, J. Schmidt, and W. T. Wenckebach, J. Magn. Reson. 77, 389 (1988)). It would be appropriate to cite the NOVEL paper, too.
3. Fig. 1a) is a sketch or diagram, but not a "phase diagram" of the setup. The term "phase diagram" has a well-defined meaning in physical chemistry. It should not be used for something different in work related to this field.
4. The non-expert reader may puzzle what you mean by low-frequency noise with respect to Fig. 4f. What you show by the dashed black line is a broad low-frequency signal due to background spins. If there are only few nearby background spins, you might observe a few lines with stochastic frequencies. You might want to allude to this very briefly in the main text and discuss it in a little more detail in the SI.
5. It may be helpful for the reader if you briefly mention in the main text that you show and discuss results on ^{14}N P1 centers in the SI.
6. You discuss line broadening by the earth magnetic field. In principle, this could be avoided by placing the setup in a μ metal box or by active compensation of the earth field. This should probably be mentioned, because it would at once improve resolution and sensitivity.
6. The caption of Figure S2 should mention that the arrows denote the three transition frequencies of the P1 center that you discuss in the main text.
7. The earth magnetic field varies a bit, but I believe that in Hefei it is closer to 0.5 G than to 0.3 G, which you quote below Eq. (S1) in the SI. This would also fit to the 1.4 GHz broadening that

you mention a few lines below.

8. It would be helpful to quote the calculation results below Eq. (S7) that you claim are consistent with experiment. They are given in the caption of Fig. S3, but not in the text.

9. Why are you testing for dependence of the 14-N transition frequencies on asymmetry of the quadrupole coupling? Is point symmetry at the N atom known for the P1 center? If there exists a rotation axis with multiplicity of at least three, the hyperfine and quadrupole tensor must both have axial symmetry. If not, I would be surprised if the hyperfine tensor would have exact axial symmetry.

10. Adding page numbers to the SI would be helpful.

11. The following typos and grammar mistakes should be corrected, as they may pose problems for a language editor:

- eigenvalues of a parameter tensor are 'principal values', not 'principle values'
- p. 3: 'phase difference before continuous driving', not 'phase difference before the continues driving'
- p. 3: 'consistent with the hyperfine coupling', not 'consisting with the hyperfine coupling'
- first paragraph of SI: 'induced by the surface dangling bonds as discussed in the main text', not 'induced by the surface dangling as discuss in the main text'

Reviewer #2 (Remarks to the Author):

The authors report on a somewhat overlooked territory in the range of nanoscale ESR techniques. They present in a very simple form how one can vary Rabi driving power to excite ESR transitions in target spins at zero magnetic field.

They show that this form of zero-field ESR allows for an almost unhindered determination of the hyperfine tensor components of P1 centers in diamond as compared to double-electron-electron-resonance form of nanoscale ESR, a field to which they also contributed greatly in sensing target spins with the nitrogen-vacancy center in diamond.

While the spectrum shown and the associated analysis are convincing, I find the manuscript lacking in several points, one of which is in fact whether this is indeed nanoscale ESR at all. Since the authors used an ensemble of NVs, detecting an ensemble of P1 centers respectively, and since no other information is given, I can only surmise that the signal arising from all NVs in the confocal spot (one micron cubed) was taken into consideration. From the low contrast of the resulting data, I also believe that this kind of measurement would not be easy to reproduce on the single spin level (NV and target spin). Therefore, the word 'nanoscale' here is somewhat problematic. I am positive that with different improvements to the experimental setup, the microwave power stability could be mitigated, but as long as that is the case, I fail to see the impact such a measurement could yield compared to the traditional ESR (induction-based) methods. If this were nanoscale ESR, then the fourth figure captures the essence of this work in a striking way, specifically figure 4c and 4d.

In general, I think more focus should be given to the impact of this method, namely page 4 and figure 4. Regarding the pulse sequence: the authors' description of it is confusing, to such extent that I am not sure someone else in the community would be able to straight-forwardly reproduce it (not impossible, but just not trivial). I believe more effort should be made in clarifying the experimental details with emphasis on the pulse scheme.

To summarize: While the manuscript definitely shows great potential, I am not completely convinced that it would have a great impact in the field. If the authors address all the points mentioned in this review and if the problematic issues (nanoscale volume, universality, analytical

model) are dealt with, it would indeed be possible to recommend it for publication in Nature Communications.

Specifically, below please find my comments for the manuscript:

- =====
01. The following was not stated clearly in the manuscript: Does one necessarily need hyperfine interaction to a nuclear spin (and a strong one at that) for this method to work? If so, then this ZF-ESR is not universal for all paramagnetic spins.
 02. If all spins give the same signal, is there a spatial feature? What is the sensing volume? What is the no. of NVs in said sensing volume? What is the no. of P1 centers in said sensing volume? How does it compare with standard (traditional) ESR?
 03. "Methods/State evolution" is not clear. Too simplistic in my humble opinion. Not only that, but I could not really understand from that section what exactly the pulse sequence used to probe the hyperfine interaction is (I could from the main text, but the "state evolution" section only confused me).
 04. This method is limited to this 400 MHz in a sense. What happens when it is a spin 7/2 with 1 GHz splitting?
 05. Have the authors really shown that these are only a few P1 centers? Can they simulate or give an analytical expression showing the difference between this so-called nanoscale sensing of few P1 centers and the sensing of many such defects?
 06. The authors write that the diamond is adhered on a CPW. From the figure (1a,1b) it looks like the opposite. What is the source of this diamond? What is its thickness? Actually, figure 1b is very misleading, since from it the reader might think that the laser is focused on a single NV center, and also that the laser focal volume does not include the P1 centers. If the authors are exciting the NVs and reading them from the backside, I can only guess it is a rather thin (100 um or less) diamond. This should be clearly stated in the text (methods).
 07. There is no mention in the text nor in the caption of figure 1 what exactly this light-blue layer on top of the diamond is. The authors do mention that different colors of arrows pertain to different spins, but do not offer a legend for this or any detail.
 08. Fig. 3a's pulse sequence is misleading or just confusing. I would give one pulse sequence and show that its amplitude is the parameter being scanned.
 09. Fig. 3c - looks like there are other very clear dips/peaks in the spectrum. What are they? Here, again, a numerical calculation of the P1 center's spectrum as probed by the NV using the ZF-ESR would have helped.
 10. All figures with measurements - what is the error? At least mention in the text if it is too small to plot, although in fig.3c it may be significant as the contrast is very poor.
 11. The contrast of the actual measurement (figure 3c) is very poor. What is the reason for this? How long did it take to acquire such a spectrum? If the measurement time is so long, did the authors take measures to account for drifts?
 12. This is, I admit, very basic, but I think there should be a derivation or analysis of equation (1) in the supp. Specifically, the difference between it and a pure (x)-only pulse sequence.
 13. Again, if ensembles were used, is this really nanoscale sensing? A typical confocal spot has a volume of 1 micron cubed (or more). If there are many NVs there and many P1 centers there, what is nanoscale? How would the signal look for real nanoscale sensing using one NV center? Would it be observable at all?
 14. Typically Omega is much larger than the hyperfine interaction. Here the comment of 400 MHz being larger than A is true for 400 MHz, but what happens when Omega is of the same order and sometimes smaller than P1's hyperfine? Is this factored in the model? Is there a model?
 15. Actually I do not see any model but just Gaussian peaks (dips) fitting. A model is kind of a minimal requirement here if the authors wish this method to mean anything.
 16. Does a radical always mean a nuclear spin next to the free electron? This relates to my first comment.
 17. The authors cite Broadway et al. as a good reference for a comparison between nanoscale ESR and ZFESR. I could not find any such comparison there. Perhaps this is a typo. If not, this reference should be more elaborate since, as I wrote, there is no such comparison there to the

best of my knowledge.

18. The equations need proofing, e.g. Eq.(8) where the z after I should be in subscript...

19. H_x is continuously driven. H_y is just a $\pi/2$ pulse, correct? Where is this treated in the state evolution section?

20. 400 MHz pulses again: Fuchs et al. actually pointed out that the RWA is not valid anymore at those strong driving conditions. How did this affect the authors' measurements? Did they use Gaussian enveloped pulses to circumvent some of the associated issues? This is not specified anywhere in the manuscript or in the supp. and so I guess not. Then I am even more puzzled that their experiment was working so beautifully.

21. There are some multiple peaks (dips) in the simulations shown in figure 4. I believe these are artefacts of the square pulses used in the simulation. If so, this should be stated. Perhaps a section in the supplementary explaining how this simulation was calculated would be of benefit to the inexperienced reader.

Reviewer #1 (Remarks to the Author):

This manuscript demonstrates zero-field EPR spectroscopy on P1 centers implanted into diamond by using nearby NV centers for sensitive detection. The experiment is based on dipole-induced polarization transfer between the NV observer and the observed P1 centers. The authors rightly argue that nanoscale EPR at zero field (or earth field) has advantages compared to such experiments in the presence of a strong magnetic field, which breaks symmetry and thus introduces orientation dependence of the transition frequencies. This is a highly significant advance in EPR spectroscopy on the nanoscale, which will probably have great influence on further development of this exciting field. It is also a very elegant use of the magnetic sensor properties of NV centers. The results and ideas are thus of sufficiently broad interest for publication in Nature Communications.

In general, the manuscript is well written, clear, and concise. The conclusions are fully supported by the experimental evidence. Yet, a few points in the discussion should be improved, there are minor mistakes, and in some places the English needs to be improved before a non-specialist can do further language editing (see below). Therefore, minor revision is required.

Reply:

We appreciate that the Reviewer recommends our work for publication in Nature Communications. As suggested by the Reviewer, we have carefully revised the manuscript and polished the language.

Details:

1. The specialist will understand from your wording that the product of $T1\rho$ with the dipole-dipole coupling between the NV center and the observed species determines whether the experiment is feasible. For a broad readership, this should be made explicit and, given that you know the dependence of $T1\rho$ on driving power, you should provide an estimate for the maximum distance between the NV center and the observe species where the experiment works.

Reply:

We have added a detailed theoretical model of this NV-P1 system in the revised SI Sec. I, and also gave an estimation of the detection area of the NV center in the revised "Methods". Specifically, the dependence of the signal on the $T1\rho$ and the dipole-dipole coupling between the NV center and the target spin is given by Eq.S8 in the revised SI. We also give an analytical estimation of the detection range (< 15 nm) of the NV center in the revised Methods.

2. It is true that you use a kind of cross-polarization technique and it may be appropriate to cite the Hartmann-Hahn paper. However, in your case polarization is transferred from a dressed species to a "bare" species, which is the type of transfer first shown in the NOVEL dynamic nuclear polarization experiment (A. Henstra, P. Dirksen, J. Schmidt, and W. T. Wenckebach, J. Magn. Reson. 77, 389 (1988)). It would be appropriate to cite the NOVEL paper, too.

Reply:

Thanks for the suggestion. The reference has been added in the revised manuscript.

3. Fig. 1a) is a sketch or diagram, but not a “phase diagram” of the setup. The term “phase diagram” has a well-defined meaning in physical chemistry. It should not be used for something different in work related to this field.

Reply:

Sorry for the mistake. It has been corrected.

4. The non-expert reader may puzzle what you mean by low-frequency noise with respect to Fig. 4f. What you show by the dashed black line is a broad low-frequency signal due to background spins. If there are only few nearby background spins, you might observe a few lines with stochastic frequencies. You might want to allude to this very briefly in the main text and discuss it in a little more detail in the SI.

Reply:

We have modified the description to clarify the low-frequency signal in the revised manuscript. Due to the dense distribution and short correlation time, the background spins always behave as a broad low-frequency signal, as observed in the Fig.S4. By further careful treatment of the diamond surface, it is indeed possible to narrow this background signal. According to the Reviewer’s suggestion, we have also discussed it in the revised SI (section II and III).

5. It may be helpful for the reader if you briefly mention in the main text that you show and discuss results on 14-N P1 centers in the SI.

Reply:

We have mentioned it in the revised manuscript (page 3, right column, first paragraph).

6. You discuss line broadening by the earth magnetic field. In principle, this could be avoided by placing the setup in a mu metal box or by active compensation of the earth field. This should probably be mentioned, because it would at once improve resolution and sensitivity.

Reply:

We have mentioned it in the revised manuscript (page 4, left column, first paragraph).

6. The caption of Figure S2 should mention that the arrows denote the three transition frequencies of the P1 center that you discuss in the main text.

Reply:

We have mentioned it in the revised SI.

7. The earth magnetic field varies a bit, but I believe that in Hefei it is closer to 0.5 G than to 0.3 G, which you quote below Eq. (S1) in the SI. This would also fit to the 1.4 GHz broadening that you mention a few lines below.

Reply:

It has been corrected in the revised SI.

8. It would be helpful to quote the calculation results below Eq. (S7) that you claim are consistent with experiment. They are given in the caption of Fig. S3, but not in the text.

Reply:

We have mentioned it in the revised SI.

9. Why are you testing for dependence of the ^{14}N transition frequencies on asymmetry of the quadrupole coupling? Is point symmetry at the N atom known for the P1 center? If there exists a rotation axis with multiplicity of at least three, the hyperfine and quadrupole tensor must both have axial symmetry. If not, I would be surprised if the hyperfine tensor would have exact axial symmetry.

Reply:

The previous purpose is to discuss a general case. However, as mentioned by the Reviewer, the asymmetry of the quadrupole coupling conflicts with the symmetry of the hyperfine coupling. Since we know that the P1 center is symmetric, we removed the part of discussion on asymmetry in the revised SI.

10. Adding page numbers to the SI would be helpful.

Reply:

We have added it in the revised SI.

11. The following typos and grammar mistakes should be corrected, as they may pose problems for a language editor:

- eigenvalues of a parameter tensor are 'principal values', not 'principle values'
- p. 3: 'phase difference before continuous driving', not 'phase difference before the continues driving'
- p. 3: 'consistent with the hyperfine coupling', not 'consisting with the hyperfine coupling'
- first paragraph of SI: 'induced by the surface dangling bonds as discussed in the main text', not 'induced by the surface dangling as discuss in the main text'

Reply:

We thank the Reviewer for pointing these mistakes. We have carefully polished the language in the revised manuscript and SI.

Reviewer #2 (Remarks to the Author):

The authors report on a somewhat overlooked territory in the range of nanoscale ESR techniques. They present in a very simple form how one can vary Rabi driving power to excite ESR transitions in target spins at zero magnetic field.

They show that this form of zero-field ESR allows for an almost unhindered determination of the hyperfine tensor components of P1 centers in diamond as compared to double-electron-electron-resonance form of nanoscale ESR, a field to which they also contributed greatly in sensing target spins with the nitrogen-vacancy center in diamond.

While the spectrum shown and the associated analysis are convincing, I find the manuscript lacking in several points, one of which is in fact whether this is indeed nanoscale ESR at all. Since

the authors used an ensemble of NVs, detecting an ensemble of P1 centers respectively, and since no other information is given, I can only surmise that the signal arising from all NVs in the confocal spot (one micron cubed) was taken into consideration. From the low contrast of the resulting data, I also believe that this kind of measurement would not be easy to reproduce on the single spin level (NV and target spin). Therefore, the word 'nanoscale' here is somewhat problematic. I am positive that with different improvements to the experimental setup, the microwave power stability could be mitigated, but as long as that is the case, I fail to see the impact such a measurement could yield compared to the traditional ESR (induction-based) methods. If this were nanoscale ESR, then the fourth figure captures the essence of this work in a striking way, specifically figure 4c and 4d.

Reply:

We thank the Reviewer for pointing out the insufficient interpretation of the nanoscale detection in our previous manuscript. To clarify this point, we have added a theoretical model in the revised SI section I, and given a quantitative estimation of the detection area (< 15 nm) of the NV center in the "Methods".

In the revised SI and "Methods", we give an analytical expression of the detected signal and show that more than 90% of the signal is contributed by the P1 centers within the 15 nm detection range, where the mean number of P1 is ~ 4 . As the mean NV spacing is ~ 135 nm (mentioned in the revised "Methods"), each NV in the confocal spot is an isolated sensor with a nanoscale detection area. Therefore, it is indeed nanoscale ESR. We also note that a recently published nanoscale ESR paper [Hall, L. T. et al. Nat. Commun. 7, 10211(2016)] also used an ensemble of NV centers to detect an ensemble of P1 centers. They claimed that the NV sensor is highly local, and the response of each NV is dominated by a few P1 centers resided in the nanoscale detection range of NV.

Secondly, the ZF-ESR measurement can be easily reproduced on single NV and single target spin level if the target spin can be put close (for example, ~ 10 nm) to the NV center. The ZF-ESR spectrum in our previous manuscript had low contrast because it was not normalized by the Rabi oscillation (the Rabi of ensembles of NVs has only $\sim 10\%$ contrast). We have modified it in the revised manuscript. Another reason is the short dephasing time (~ 0.1 us) of P1 centers (according to Eq. S21 in the revised SI). The short dephasing time is induced by the dense bath spins from the ions implantation. For a separable P1 or other defects, the bath is much purer, and the dephasing time is estimated to be \sim us [PRB 87, 195414 (2013)]. Thus an order of increase of the signal contrast is expected. Besides, a single NV has $\sim 40\%$ Rabi contrast. Therefore, the reproduction on the single spin level may be potentially more efficient.

Our purpose of using an ensemble of NVs rather than single NV is to make the P1 close to the NV, for which high-dose ions implantation is required. It leads to high density of NVs, which cannot be resolved by the confocal microscope. However, it does not affect the nanoscale detection area of the NV center.

In general, I think more focus should be given to the impact of this method, namely page 4 and figure 4. Regarding the pulse sequence: the authors' description of it is confusing, to such extent that I am not sure someone else in the community would be able to straight-forwardly reproduce it (not impossible, but just not trivial). I believe more effort should be made in clarifying the experimental details with emphasis on the pulse scheme.

Reply:

We have modified the paragraph related to figure 4 to clarify the significant advantage of our method. By using NV centers, the resolution of ESR has been significantly improve from mm~ μ m scale to nanoscale. Besides, this nanoscale ZF-ESR removes the two serious obstacles of the applications of nanoscale ESR, i.e., the spectrum dispersion induced by the random orientations of target spins and the significant background signal. Furthermore, different from the induction-based ESR, the sensitivity of NV does not depend on the magnetic field or the operation frequency, and thus the sensitivity of nanoscale ZF-ESR is comparable and even potentially better than nanoscale ESR.

In the revised "Methods", we have given a detailed description of the pulse sequence.

To summarize: While the manuscript definitely shows great potential, I am not completely convinced that it would have a great impact in the field. If the authors address all the points mentioned in this review and if the problematic issues (nanoscale volume, universality, analytical model) are dealt with, it would indeed be possible to recommend it for publication in Nature Communications.

Reply:

We appreciate the Reviewer's patient comments. In the revised manuscript and SI, we have addressed all the concerns given by the Reviewer.

Specifically, below please find my comments for the manuscript:

=====

01. The following was not stated clearly in the manuscript: Does one necessarily need hyperfine interaction to a nuclear spin (and a strong one at that) for this method to work? If so, then this ZF-ESR is not universal for all paramagnetic spins.

Reply:

We have emphasized in the last paragraph (also mentioned in the second paragraph, right column, page 2) that the key of our method is the direct measurement of the energy level structure of paramagnetic spins. The interaction, which induces energy level splitting, can be any forms, including the electron-nuclear hyperfine interaction, electron-electron fine interaction, and even many body interactions, etc.

The detection range of the energy level splitting is determined by the Rabi frequency. The lower bound of Rabi frequency is limited by the transverse relaxation rate of the NV center, which is ~ MHz. While the upper bound of Rabi frequency is limited by the zero-field splitting of the NV center, which is ~ GHz (detail in the reply of comment #4). For some isolated radicals without any intrinsic interactions, it is indeed difficult to detect them by our method as the energy splitting is near zero. However, we believe that this frequency detection range (MHz-GHz) can cover most of the usual paramagnetic radicals and complexes.

02. If all spins give the same signal, is there a spatial feature? What is the sensing volume? What is the no. of NVs in said sensing volume? What is the no. of P1 centers in said sensing volume? How does it compare with standard (traditional) ESR?

Reply:

We apologize for the unclear statement in the previous version. The resonance frequencies (i.e. signal position) of all the spin are the same, but the coupling rates with the NV (i.e. signal intensity) are spatially dependent. The statement has been corrected and a detail description is given in the revised SI.

As mentioned above, the detection range is less than 15 nm with about four P1 centers residing in the detection area of each NV, the mean NV spacing is ~ 135 nm with ~ 100 NVs residing in the confocal spot. Here we note again that the NV is a local sensor, and all the NV sensors are working independently. There is no principal limitation for further single-NV applications.

For traditional ESR, the best spatial resolution is \sim micron (The traditional ZF-ESR usually need a sample with size of \sim cm). While for our NV-based ZF-ESR, the spatial resolution is ~ 10 nm, it is indeed a significant improvement.

03. "Methods/State evolution" is not clear. Too simplistic in my humble opinion. Not only that, but I could not really understand from that section what exactly the pulse sequence used to probe the hyperfine interaction is (I could from the main text, but the "state evolution" section only confused me).

Reply:

We apologize for the unclearness here. The "Methods" has been rewritten. The Section "Spin locking in zero field" gives a detailed description of the pulse sequence.

04. This method is limited to this 400 MHz in a sense. What happens when it is a spin 7/2 with 1 GHz splitting?

Reply:

The frequency detection range of our method is currently limited by the microwave amplifier (with saturated output power of 16 W), and further limited by the energy level splitting of the NV (i.e. zero-field splitting, 2.87GHz). The Reviewer also mention the work [Fuchs et al. Science 326,1520–1522 (2009)]. They pointed out that the RWA is not valid anymore at strong driving conditions. Here the 'strong' is relative, as they also point out the Bloch-Siegert shift induced by the driving field is $\sim H_1^2/4H_0^2$. For their experiment, $H_0 \sim 0.49$ GHz and the Rabi oscillation remained clear up to ~ 200 MHz by using shaped pulses. Thus for the situation in our work, where $H_0 \sim 2.87$ GHz, the driving power up to ~ 1.2 GHz is possible. We also note that this limit can be further promoted by employing circularly polarized microwave [PRB 76, 165205 (2007)]. We emphasize again that our method have a broad frequency detection range (MHz-GHz), yet it is not applicable for some particular spins with very large splitting.

05. Have the authors really shown that these are only a few P1 centers? Can they simulate or give an analytical expression showing the difference between this so-called nanoscale sensing of few P1 centers and the sensing of many such defects?

Reply:

As suggested by the Reviewer, we have given a theoretical model of the NV-P1 system and detailed calculations of the signal in the revised SI (section I) and "Methods". As mention above, the NV center is a natural local sensor. According to our calculations, even if the NV center is surrounded by numerous P1 centers, the signal of all the P1 centers outside the detection area is

illegible. More than 90% of the signal is contributed by about four P1 centers residing in the nanoscale detection area.

06. The authors write that the diamond is adhered on a CPW. From the figure (1a,1b) it looks like the opposite. What is the source of this diamond? What is its thickness? Actually, figure 1b is very misleading, since from it the reader might think that the laser is focused on a single NV center, and also that the laser focal volume does not include the P1 centers. If the authors are exciting the NVs and reading them from the backside, I can only guess it is a rather thin (100 um or less) diamond. This should be clearly stated in the text (methods).

Reply:

Both the figure and its caption are modified in the revised manuscript. The diamond is indeed adhered on a CPW, as the CPW is much larger than the diamond. Here is a sketch of the setup.

The detail information of the diamond is given in the revised "Methods". Specifically, the diamond is obtained commercially with a thickness of 500 μm . The working distance of the objective we used is 1.5-2.2 mm.

07. There is no mention in the text nor in the caption of figure 1 what exactly this light-blue layer on top of the diamond is. The authors do mention that different colors of arrows pertain to different spins, but do not offer a legend for this or any detail.

Reply:

It has been modified in the revised version. The green arrows in the light-blue layer denote the outer target spins.

08. Fig. 3a's pulse sequence is misleading or just confusing. I would give one pulse sequence and show that its amplitude is the parameter being scanned.

Reply:

It has been modified according the Reviewer's suggestion.

09. Fig. 3c - looks like there are other very clear dips/peaks in the spectrum. What are they? Here, again, a numerical calculation of the P1 center's spectrum as probed by the NV using the ZF-ESR would have helped.

Reply:

As the diamond we used is implanted with high dose of ions, there are numerous vacancy defects in the diamond, which might be the source of these peaks. We also mentioned it in the revised manuscript. The calculated spectrum is given in the revised SI.

10. All figures with measurements - what is the error? At least mention in the text if it is too small to plot, although in fig.3c it may be significant as the contrast is very poor.

Reply:

It has been modified in the revised version. The SNR of fig.3c is ~ 10 .

11. The contrast of the actual measurement (figure 3c) is very poor. What is the reason for this? How long did it take to acquire such a spectrum? If the measurement time is so long, did the authors take measures to account for drifts?

Reply:

As mention above, the data in the previous version were not normalized by the Rabi. After normalization, the signal contrast is ~3%. We have modified it in the revised manuscript. Another reason is the short dephasing time (~0.1 us) of P1 centers (according to Eq. S21 in the revised SI).

The measurement takes for ~4 hours. As mentioned in the SI section IV.A, the measurement is divided into many rounds, each lasts for ~3 minutes. Among the rounds, a Rabi oscillation is measured to record and then calibrate the pulse sequence.

12. This is, I admit, very basic, but I think there should be a derivation or analysis of equation (1) in the supp. Specifically, the difference between it and a pure (x)-only pulse sequence.

Reply:

A derivation is given in the revise “Methods” (section “Spin locking in zero field”).

13. Again, if ensembles were used, is this really nanoscale sensing? A typical confocal spot has a volume of 1 micron cubed (or more). If there are many NVs there and many P1 centers there, what is nanoscale? How would the signal look for real nanoscale sensing using one NV center? Would it be observable at all?

Reply:

As mentioned above, the detailed calculations in the revised version confirm the detection area is nanoscale. Because the NV sensors are working independently, the signal of many NVs is the average rather than the sum of the signals of all the detected NVs. If a single NV is used, the signal contrast should be much higher due to the higher Rabi contrast and the longer dephasing time.

14. Typically Omega is much larger than the hyperfine interaction. Here the comment of 400 MHz being larger than A is true for 400 MHz, but what happens when Omega is of the same order and sometimes smaller than P1's hyperfine? Is this factored in the model? Is there a model?

Reply:

Here Omega is adjustable, and the ZF-ESR spectrum is acquired by sweeping Omega. The signal appears only when Omega matches the energy level splitting of P1, which is determined by the P1's hyperfine.

By the way, in the previous Methods/State evolution, we stated that $\Omega \gg A$, which is misleading. There A means the hyperfine between NV electron and nuclear spins rather than the P1's hyperfine, which is just 3 MHz. We have modified the description in the revised “Methods” to avoid this misunderstanding. When Omega is of the same order or smaller than 3 MHz, an obvious offresonance is expected. However, we can polarize the nuclear spin by SWAP gate [L. Jiang, et al. Science 326, 267 (2009)] before the measurement to remove this offresonance. Therefore, the hyperfine of the NV is not a limitation of our method. The lower bound of Omega is limited by the dephasing time of the NV center.

15. Actually I do not see any model but just Gaussian peaks (dips) fitting. A model is kind of a minimal requirement here if the authors wish this method to mean anything.

Reply:

We have added a model in the revised SI section I.

16. Does a radical always mean a nuclear spin next to the free electron? This relates to my first comment.

Reply:

The previous description of radical is less strict. We have modified it in the revised manuscript. We just take a radical with hyperfine interaction as an example, in order to explain how the energy level structure is resolved by the ZF-ESR.

17. The authors cite Broadway et al. as a good reference for a comparison between nanoscale ESR and ZFESR. I could not find any such comparison there. Perhaps this is a typo. If not, this reference should be more elaborate since, as I wrote, there is no such comparison there to the best of my knowledge.

Reply:

Our purpose is to compare the sensitivity of the cross-polarization measurement (ZF-ESR) and the interference measurement (ESR), which is discussed by Broadway et al. Indeed, the reader may be confused. We have modified the description of this comparison in the revised manuscript to clarify that nanoscale ZF-ESR has comparable and potentially better sensitivity than nanoscale ESR. Specifically, the nanoscale detection is based on dipole-dipole coupling between the probe and the target and limited by the coherence properties of the probe. For DEER measurement, the sensitivity is limited by the T2 of the NV center. While for ZF-ESR measurement, the sensitivity is limited by the T1rho of the NV center, which is usually much larger than T2. So the sensitivity of our nanoscale ZF-ESR is comparable with and even potentially better than the nanoscale ESR.

18. The equations need proofing, e.g. Eq.(8) where the z after l should be in subscript...

Reply:

Sorry for the mistake. We have carefully proofed both the manuscript and the SI.

19. Hx is continuously driven. Hy is just a pi/2 pulse, correct? Where is this treated in the state evolution section?

Reply:

We have given a detail description of the pulse sequence in the revise "Methods", section "spin locking in zero field". For brief, the pi/2 Hy pulse prepares the eigenstate of Hx, and then Hx is continuously driven to lock this state.

20. 400 MHz pulses again: Fuchs et al. actually pointed out that the RWA is not valid anymore at those strong driving conditions. How did this affect the authors' measurements? Did they use Gaussian enveloped pulses to circumvent some of the associated issues? This is not specified

anywhere in the manuscript or in the supp. and so I guess not. Then I am even more puzzled that their experiment was working so beautifully.

Reply:

As mention above, the words “strong” is define by driving power/energy splitting. The energy splitting in Fuchs et al. is 0.49 GHz, while it is 2.87 GHz in our work. Specifically, the energy shift (i.e. Bloch-Siegert shift) induced by the strong driving filed at 400 MHz is just $\sim 0.5\%$ in our work.

We do not use Gaussian enveloped pulses in this work, but it may be further used if a larger Omega is required, as mentioned in the reply of 4th comment.

21. There are some multiple peaks (dips) in the simulations shown in figure 4. I believe these are artefacts of the square pulses used in the simulation. If so, this should be stated. Perhaps a section in the supplementary explaining how this simulation was calculated would be of benefit to the inexperienced reader.

Reply:

The multiple peaks are indeed the artefacts of the square pulses. In the revised version, we also take care of the dephasing of target spins, and then these artefacts will disappear. The theoretical model of ZF-ESR is given in the revised SI. The simulation method of the DEER spectrum is not included, which can be found in the DEER papers we cited.

REVIEWERS' COMMENTS:

Reviewer #1 (Remarks to the Author):

I am satisfied with the revision by the authors and I do think that this qualifies as nanoscale EPR. My suggestion is to publish this manuscript in its current form.

Reviewer #2 (Remarks to the Author):

I am now much more pleased with the manuscript. The authors have successfully addressed all of my concerns. I recommend publication in Nature Communications.

I have two comments and four corrections (the authors may want to go over the manuscript one more time to find more such possible typographical errors):

1. In the abstract, there should be "the" before "nitrogen-vacancy (NV) center in diamond".
2. abstract: "door of practical"  "door for practical".
3. In page three, second column, they write: "...or the P1 center will destruct the energy". The word "destruct" should be replaced, of course, with "destroy".
4. Methods: Perkin Elemer  Perkin Elmer
5. Perhaps the authors should specify the lower limit frequency of their method (bounded by the NV's T_2^*)
6. How will dipole-dipole interactions with nearby (^{13}C) __nuclear__ spins will manifest in the ZF-ESR spectra? Could they explain the unknown dips in Fig.3c?

Reviewer #1 (Remarks to the Author):

I am satisfied with the revision by the authors and I do think that this qualifies as nanoscale EPR. My suggestion is to publish this manuscript in its current form.

Reply:

We appreciate that the Reviewer recommends our work for publication in Nature Communications.

Reviewer #2 (Remarks to the Author):

I am now much more pleased with the manuscript. The authors have successfully addressed all of my concerns. I recommend publication in Nature Communications.

Reply:

We appreciate that the Reviewer recommends our work for publication in Nature Communications.

I have two comments and four corrections (the authors may want to go over the manuscript one more time to find more such possible typographical errors):

1. In the abstract, there should be "the" before "nitrogen-vacancy (NV) center in diamond".

2. abstract: "door of practical"  "door for practical".

3. In page three, second column, they write: "...or the P1 center will destruct the energy". The word "destruct" should be replaced, of course, with "destroy".

4. Methods: Perkin Elemer  Perkin Elmer

Reply:

We thank the Reviewer for pointing these mistakes, which has been corrected in the revised manuscript.

5. Perhaps the authors should specify the lower limit frequency of their method (bounded by the NV's T2*)

Reply:

We have mentioned it in the revised manuscript.

6. How will dipole-dipole interactions with nearby (^{13}C) __nuclear__ spins will manifest in the ZF-ESR spectra? Could they explain the unknown dips in Fig.3c?

Reply:

The hyperfine interaction (including the dipole-dipole interaction and the Fermi contact interaction) between the P1 electron spin and a nearby ^{13}C nuclear spin will induce an extra splitting of the resonance lines, and thus the positions of the dips will change. Considering the

large difference between the positions of the P1 dips and the extra dips, only the interaction with the nearest ^{13}C nuclear spins may possible induce such large deviations. However, as the natural abundance of ^{13}C for the diamond used in this work is just 1.1%, the average signal of those P1 centers with a nearest ^{13}C is illegible. So we think a more possible explanation of the extra dips is the existence of other kinds of defects.